# Display of Microbial Glucose Dehydrogenase and Cholesterol Oxidase on the Yeast Cell Surface for the Detection of Blood Biochemical Parameters

**DOI:** 10.3390/bios11010013

**Published:** 2020-12-30

**Authors:** Shiyao Zhao, Dong Guo, Quanchao Zhu, Weiwang Dou, Wenjun Guan

**Affiliations:** 1Institute of Pharmaceutical Biotechnology and the Children’s Hospital, Zhejiang University School of Medicine, Hangzhou 310012, China; zhaoshiyao@zju.edu.cn (S.Z.); 11818296@zju.edu.cn (Q.Z.); douww@zju.edu.cn (W.D.); 2College of Pharmaceutical Sciences, Zhejiang University, Hangzhou 310012, China; Y215190015@zju.edu.cn

**Keywords:** whole-cell biosensor, yeast surface display, cholesterol oxidase, glucose dehydrogenase, electrochemical detection

## Abstract

High levels of blood glucose are always associated with numerous complications including cholesterol abnormalities. Therefore, it is important to simultaneously monitor blood glucose and cholesterol levels in patients with diabetes during the management of chronic diseases. In this study, a glucose dehydrogenase from *Aspergillus oryzae* TI and a cholesterol oxidase from *Chromobacterium* sp. DS-1 were displayed on the surface of *Saccharomyces cerevisiae*, respectively, using the yeast surface display system at a high copy number. In addition, two whole-cell biosensors were constructed through the immobilization of the above yeast cells on electrodes, for electrochemical detection of glucose and cholesterol. The assay time was 8.5 s for the glucose biosensors and 30 s for the cholesterol biosensors. Under optimal conditions, the cholesterol biosensor exhibited a linear range from 2 to 6 mmol·L^−1^. The glucose biosensor responded efficiently to the presence of glucose at a concentration range of 20–600 mg·dL^−1^ (1.4–33.3 mmol·L^−1^) and showed excellent anti-xylose interference properties. Both biosensors exhibited good performance at room temperature and remained stable over a three-week storage period.

## 1. Introduction

Diabetes is currently a global epidemic with over 400 million cases and the prevalence of which is rapidly increasing [1,2]. The disease is associated with numerous complications, especially cardiovascular diseases. Indices of blood cholesterol are often used as the thresholds for risk assessment and guides to therapy [3,4,5]. Therefore, effective monitoring of blood glucose and cholesterol plays a crucial role in the management of diabetes. Currently, large biochemical analyzers are the mainstream methods of detecting diabetes related blood parameters in clinical practice [6,7,8]. However, the approach requires specialized instruments and complicated experimental procedures [9] although it is highly reliable and gives high precision. With increasing demand for point-of-care testing (POCT), biosensors have been proposed as an attractive alternative as they are quick, convenient and economically feasible [10]. Among them, the optical and electrochemical based POCT biosensors gradually become the leader choices [11]. Because significant turbidity of most real samples and strong light from environment always bring errors, the detection accuracy of optical biosensors is easily interfered [12]. Hence, electrochemical biosensors are more favorable in practical applications [13]. At present, the most well-known brands of glucose POCT biosensors based on electrochemical technology on the market are Roche, Johnson & Johnson, Bayer, Abbott, et al.

Enzyme-based biosensors represent the most popular group of electrochemical biosensors currently available [14]. Notably, glucose oxidase or glucose dehydrogenase is the most widely used enzyme for glucose detection [15,16]. Glucose dehydrogenase forms a complex with cofactors such as flavin adenine dinucleotide (FAD), nicotinamide adenine dinucleotide (NAD), or pyrroloquinoline quinone (PQQ) [17]. Compared with glucose oxidase and other kinds of glucose dehydrogenases, the FAD-dependent glucose dehydrogenase, predominantly found in *Aspergillus*, has obvious sensing advantages due to its favorable substrate specificity and insensitivity to oxygen [18]. On the other hand, cholesterol oxidase, which is a FAD-dependent oxidoreductase derived from bacteria, is most widely utilized for the detection of cholesterol [19]. Nonetheless, the high cost of purification and poor stability of the free enzymes still present a challenge for the large-scale production and practical application of these enzyme-based biosensors [20]. Therefore, electrochemical whole-cell biosensors based on the surface display technology have been the focus of several studies since they may provide an effective way of overcoming the challenges associated with their enzyme-based counterparts [21].

Surface display is a powerful technology that allows the presentation of multiple proteins on the surface of microbes such as bacteria and yeast [22]. For instance, previous studies displayed a *Bacillus subtilis* derived NAD-dependent glucose dehydrogenase and its mutants on the surface of *Escherichia coli* using the ice nucleation protein (INP) as an anchoring motif [23,24]. However, no report currently exists on microbial cell-surface display of cholesterol oxidase. The yeast surface display (YSD) system is more advantageous in displaying complex eukaryotic proteins which require post-translational modification or have high molecular mass [25]. The most common yeast display system, pioneered by Boder and Wittrup, employs the Aga1 and Aga2 subunits of a-agglutinin to anchor the target enzymes onto the cell wall of *Saccharomyces cerevisiae* [26]. The anchoring of enzymes on the yeast cell surface allows for direct enzymatic reaction with substrates without the need for purification of enzymes and this can significantly reduce the cost of preparation as well as application of enzyme-based biosensors [27]. Moreover, the surface of yeast cells provides a biocompatible microenvironment that helps in maintaining the stability of enzymes [28]. The above advantages indicate that displaying glucose dehydrogenase and cholesterol oxidase on the surface of yeast cells may potentially be useful in the development of electrochemical biosensing platforms for the detection of blood glucose and cholesterol.

In this study, a FAD-dependent glucose dehydrogenase gene from *Aspergillus oryzae* TI [29] and a cholesterol oxidase gene from *Chromobacterium* sp. DS-1 [30] were cloned and the corresponding enzymes were displayed on the surface of *S. cerevisiae* through the Aga1-Aga2 system, respectively. Afterwards, the enzyme-displayed yeast cells were immobilized onto electrodes to construct electrochemical biosensors for the detection of glucose and cholesterol. The catalytic activity of the surface-displayed enzymes was measured carefully and the performance of the two related biosensors was evaluated subsequently. The results demonstrated that this strategy had the advantages of simplicity, economic feasibility, and stability in the application of glucose and cholesterol biosensors.

## 2. Materials and Methods

### 2.1. Strains, Media and Reagents

The strains used in this study are described in Appendix A. *E. coli* TG1 [31] was used for recombinant DNA manipulations and was cultured in Luria-Bertani (LB) medium at 37 °C. *S. cerevisiae* EBY100 [26] was obtained from Ziyun Biotech (Hangzhou, Zhejiang, China) and used for yeast cell surface display. The *S. cerevisiae* EBY100 was first grown in a seed medium containing 0.67% yeast nitrogen base, 0.5% casamino acid and 2% glucose before being transferred into the induction medium for surface display of the enzymes. The composition of induction medium was the same as that of the seed medium only that glucose was replaced with galactose. The protein expression was induced at 20 °C with continuous shaking at 220 rpm. Restriction enzymes were purchased from Takara Bio (Shiga, Japan) while all the other biochemical reagents were of at least analytical grade and purchased from Merck (Darmstadt, Germany), Sangon Biotech (Shanghai, China), Aladdin (Shanghai, China) or Sinopharm Chemical Reagent (Shanghai, China).

### 2.2. Construction of Vectors

All the vectors used in this study are listed in Appendix A. The cholesterol oxidase gene derived from *Chromobacterium* sp. DS-1 (named *CHO1*, Sequence ID: AB456533.1) and glucose dehydrogenase gene derived from *Aspergillus oryzae* TI (named *GDH1*, Sequence ID: XM_002372558.1) were codon optimized and synthesized by Generay Biotech (Shanghai, China). To construct the vectors pYD1-CHO1 and pYD1-GDH1, the synthetic genes were digested with *Bam*HI and *Eco*RI before being ligated into the multiple cloning site of vector pYD1 [32]. In addition, the extended linkers were prepared by annealing synthetic complementary single-stranded DNAs of the target sequences followed by PCR. The linkers added one or two proline-alanine-serine (PAS) sequences (ASPAAPAPASPAAPAPSAPA) to the original GS linker (GGGGSGGGGSGGGGS) in pYD1-CHO1. Thereafter, the amplification products were cloned into the *Hin*dIII and *Bam*HI sites of vector pYD1-CHO1 to generate pYD1-CHO1-PASx1 and pYD1-CHO1-PASx2. All the resulting vectors were verified through PCR and sequencing.

### 2.3. Freeze-Drying of Yeast Cells

After induction, the enzyme-displayed yeast cells harboring vector pYD1-CHO1 or pYD1-GDH1 were harvested through centrifugation, washed with 1 × phosphate buffered saline (PBS, pH 7.4) and resuspended in the cryoprotectant buffer which consisted of 1 × PBS and 5% glycerol. Afterwards, the cell suspensions were freeze-dried using a freeze dryer (LGJ-10, Henan Brother Equipment Co., Ltd., Zhengzhou, China). The freeze-dried cell samples were then stored at 4 °C for later use.

### 2.4. Enzyme Activity Assays

Glucose dehydrogenase activity of the Gdh1-displayed yeast cells was assessed using 2,6-dichlorophenol-indophenol (DCPIP) and phenazine methosulfate (PMS) according to a previously published protocol [33]. Briefly, the working solution was configured, and the final concentration of glucose was 201 mmol·L^−1^. 1.5 mL of the working solution was equilibrated at 37 °C for about 5 min, and then the 20 OD_600_ freeze-dried cell samples were resuspended in this solution to initiate the reaction. During a 5-min reaction, a decrease in the optical density of the supernatant was measured at 600 nm using a spectrophotometer, with water as the reference. Finally, change in absorbance per minute was used to calculate enzyme activity.

Cholesterol oxidase activity of the Cho1-displayed yeast cells was determined through the oxidative coupling of 4-aminoantipyrine and phenol as previously described [34]. Configuration of the working solution was performed according to this method and the final concentration of cholesterol in the solution was 0.89 mmol·L^−1^. After incubating the 1.5 mL working solution at 37 °C for 5 min, the 20 OD_600_ freeze-dried cell samples were resuspended with this solution to start the reaction. The amount of H_2_O_2_ catalyzed by cholesterol oxidase was then calculated by measuring the increase in OD_500_ of the supernatant per minute, to define the value of enzyme activity.

### 2.5. Fabrication of the Whole-Cell Biosensors

The biosensor used in this study consisted of a two-electrode system made from carbon paste or gold. The carbon paste-based two-electrode strips were formed by successively printing silver ink, carbon ink and insulating ink on a polyethylene terephthalate (PET) material through the screen-printing technique. The gold two-electrode strips were purchased from Jinhong Technology (Beijing, China). The size of the reaction chamber for each two-electrode strip was 5 mm long, 1.95 mm wide and 0.125 mm high. The 20 OD_600_ freeze-dried YSD cells were resuspended in 50 µL of the respective buffers before adding 0.05 g of FAD to mix. The YSD cell solution and electrochemical solution which contained microcrystalline cellulose, polyvinylpyrrolidone, octyl polyethylene glycol phenyl ether, phenazine ethosulfate, trehalose and the electron mediator (potassium ferricyanide or hexaammineruthenium (III) chloride), were mixed in a ratio of 1:4. The reaction chamber of each two-electrode strip was spotted with 1 µL of the above mixed solution and dried at 30 °C for 20 min. In theory, each resulting biosensor contained approximately 0.08 OD_600_ YSD cells. The mechanism of detection is described in Appendix A, and a schematic illustration of the reaction chamber is shown in Appendix A. Performance of the screen-printed carbon electrodes was evaluated before the formal testing (Appendix A).

### 2.6. Preparation of Whole Blood Samples

Whole blood samples were collected into sodium heparin tubes to prevent hemolysis. The initial hematocrit (HCT) was determined by testing the volume of plasma and red blood cells after centrifugation before adjusting it to 42% by adding or removing plasma. The initial concentration of glucose in whole blood samples was measured using a YSI glucose analyzer (YSI2300, YSI Life Sciences, Yellow Springs, OH, USA). Glucose solutions were supplemented to obtain the desired concentrations in whole blood samples.

### 2.7. Electrochemical Measurements

All electrochemical measurements were performed using an electrochemical workstation (CHI660E, CH Instruments, Shanghai, China) or a portable electrochemical monitor (305A, Jiangsu Yuyue Medical Instruments Co., Ltd., Jiangsu, China). Amperometric method, in which a constant potential was applied to the working electrode and the current was measured after a certain reaction time, was used in this work. For the glucose biosensor, a DC voltage of 0.3 V was applied on the working electrode for 8.5 s and a final current value at 8.5 s was read for data analysis. In addition, for the cholesterol biosensor, a DC voltage of 0.3 V was applied on the working electrode for 30 s and a final current value at 30 s was used for data analysis.

## 3. Results

### 3.1. Surface Display of Glucose Dehydrogenase

Previous reports suggested that the glucose dehydrogenase derived from *Aspergillus oryzae* TI is a FAD-dependent oxidoreductase, which has the characteristics of high substrate specificity against glucose, excellent thermal stability and is not affected by dissolved oxygen [29]. This enzyme is composed of 593 amino acids, including a signal peptide ranging from 1–22 amino acids at the N-terminus [35]. To display Gdh1 on the surface of yeast, the signal peptide truncated gene sequence (1713 bp) of Gdh1 was synthesized according to the codon preference in *S. cerevisiae* and introduced into the multiple cloning site of vector pYD1. The resulting vector pYD1-GDH1 carrying the *AGA2*-*GDH1* fusion gene (Figure 1a) was then transformed into *S. cerevisiae* EBY100, which harbored Aga1 as a cell wall anchoring motif, to generate the G1 strain (Figure 1b). The expression of *AGA1* as well as *AGA2* was driven by the galactose-inducible *GAL1* promoter. The G1 strain was cultured in a 2% glucose medium to the mid-logarithmic growth phase, and then was transferred to a 2% galactose medium to induce the persistent expression of Gdh1. Afterwards, the G1 yeast cells were collected at the end of fermentation and freeze dried.

Maximum catalytic activity (0.7 U per OD_600_ cells) of the surface-displayed Gdh1 was observed after 48 h of galactose induction (Figure 2a). Therefore, 48 h was selected as the optimal induction time for the G1 strain and used for subsequent experiments. The effect of temperature, pH and storage time on the activity of Gdh1 was also evaluated. The results showed that Gdh1 was active from 10 to 60 °C and the activity reached a peak at 30 °C. In addition, the enzyme retained over 95% of its activity from 20 °C to 40 °C, indicating that it had good thermal stability and could be applied over a broad range of temperature (Figure 2b). The optimum pH for Gdh1 activity was observed to be 7.0. More than 75% of Gdh1 activity was maintained within the pH range of 6.5–7.5 (Figure 2c). However, the storage stability of the surface-displayed Gdh1 appeared to be limited. Activity of the enzyme began to decrease following storage at 4 °C for one week and retained just 20% of the initial activity after 3 weeks of storage (Figure 2d). This implied that the subsequent immobilization process should be performed as soon as possible after obtaining the freeze-dried G1 strain cells to avoid the loss of enzyme activity.

### 3.2. Development of an Electrochemical Glucose Biosensor Based on the Surface-Displayed Gdh1

To develop a whole-cell glucose biosensor, the freeze-dried G1 yeast cells (0.08 OD_600_ cells per strip) with different artificial electron mediators and buffer solutions, were immobilized on the surface of screen-printed carbon electrodes. Electrochemical detection was then performed. Although the glucose biosensors exhibited a similar slope of the concentration-response current curve when different electron mediators were used, the glucose biosensor using hexaammineruthenium (III) chloride as electron mediator presented lower background current than that of potassium ferricyanide (Figure 3a). In addition, the impact of phosphate buffer, heppso-malic acid buffer and fumaric acid buffer was evaluated to examine which one of them was better suited for the glucose biosensor. The results showed that the biosensor based on the 0.1 mol·L^−1^ phosphate buffer (pH 7.0) system had a greater response current at a higher glucose concentration (Figure 3b). Consequently, under optimized electrode conditions, the biosensor was employed for the detection of glucose with successive additions. The linear detection range of the glucose biosensor was 20–600 mg·dL^−1^ (1.4–33.3 mmol·L^−1^) and the testing sensitivity was shown to be 25 mg·dL^−1^ at the concentration range of 20 to 600 mg·dL^−1^ (Figure 3c,d).

Xylose is the most common interferent to biosensors using glucose dehydrogenase. Therefore, the study examined the selectivity of the optimized glucose biosensor by adding extra xylose into the glucose substrate solution. Addition of xylose had almost no impact on the biosensor developed by this study. However, the biosensor immobilized with commercial FAD-dependent glucose dehydrogenase (581 U·mg^−1^, Code: GLD-361, Toyobo Co., Ltd., Osaka, Japan) was affected substantially. 189% and 29% of additional interference current was generated in 70 mg·dL^−1^ and 300 mg·dL^−1^ of glucose substrate solution after adding 90 mg·dL^−1^ of xylose respectively (Figure 3e). This indicated that the optimized glucose biosensor had a good ability to get rid of xylose interference and great potential for practical application.

The biosensors were stored at room temperature for 21 days and changes in current responses towards 280 mg·dL^−1^ of glucose were measured. The findings revealed that the biosensors retained 79% of the initial response current after storage for 21 days (Figure 3f). Furthermore, the biosensor was used to determine the concentration of glucose in whole blood samples to explore the possibility of clinical application. The results showed that the linear response range of the biosensor to glucose in whole blood samples was 20–600 mg·dL^−1^ (Figure 3g). Although at present the accuracy of the glucose biosensor did not completely meet the requirements of the standard [36], it exhibited the potential of detecting real samples (Appendix A, Appendix A).

### 3.3. Surface Display of Cholesterol Oxidase

The strategy for developing a glucose biosensor based on YSD cells proved to be feasible and showed potential for practical application in monitoring glucose levels in whole blood samples. Therefore, the study used a similar approach for the detection of cholesterol. Previous studies reported that the cholesterol oxidase derived from *Chromobacterium* sp. DS-1 had favorable protein structural characteristics [37], excellent stability [30] and high enzyme activity [38]. This enzyme consists of 584 amino acids with the first 44 being a signal peptide [38]. In the present study, the codon optimized gene sequence (1620 bp) encoding Cho1 without the signal peptide was synthesized and the resulting vector pYD1-CHO1 was constructed (Figure 1a). Thereafter, pYD1-CHO1 was transformed into *S. cerevisiae* EBY100 to generate the C1 strain, which successfully displayed Cho1 on the surface of yeast (Figure 1b). It was observed that the C1 strain had a significantly lower growth rate compared with the G1 strain or the P1 strain containing the empty vector pYD1. This implied that displaying Cho1 on the surface might impose a growth burden to the cells (Appendix A).

The optimal induction time for the C1 strain was shown to be 36 h, at which enzyme activity reached the maximum value of 3.4 × 10^−3^ U per 20 OD_600_ freeze-dried cells in vitro (Figure 4a). The optimum reaction temperature for the surface-displayed Cho1 was also found to be 30 °C. More than 60% of its activity occurred between 20 °C and 40 °C, while about 18% activity was retained at 60 °C (Figure 4b). Cho1 had an optimum pH of 6.5 and over 50% of its activity occurred in the pH range of 5.5 to 7.5 (Figure 4c). Afterwards, the freeze-dried C1 strain cells were stored at 4 °C and their residual activity determined intermittently within a one-month period, to investigate the stability of Cho1 in vitro. No significant decrease in activity was observed and approximately 95% of original enzyme activity could still be detected after the 4-week storage period. This suggested that the C1 strain cells had good storage stability (Figure 4d).

The maximum activity of Cho1 was determined as 3.4 × 10^−3^ U per 20 OD_600_ freeze-dried yeast cells and this was almost two orders of magnitude lower than that of Gdh1. This implied that the activity of surface-displayed Cho1 had a limited potential to meet the requirements for practical application. Therefore, an attempt was made to improve its activity by increasing the length of linker sequence, which was located between *AGA2* and *CHO1* in the pYD1-CHO1 vector. Based on previous reports, the original GS linker was modified by adding one or two PAS linkers, which consisted of 20 amino acids including proline, alanine and serine, with the advantages of being hydrophilic, uncharged and structureless [39,40] (Figure 4e). The resulting vector pYD1-CHO1-PASx1 or pYD1-CHO1-PASx2 was then transformed into *S. cerevisiae* EBY100 to generate the C2 or C3 strains, respectively. The results showed that the displayed Cho1 with a PAS × 1 + GS linker had a 19% increase in activity, while the one with a PAS × 2 + GS linker exhibited a 62% increase, which was 5.5 × 10^−3^ U per 20 OD_600_ freeze-dried cells (Figure 4f). Owing to the longer linkers, the displayed enzyme obtained a greater distance to leave the cell surface, hence had enough conformational space, resulting in a decrease in the loss of enzymatic activity.

### 3.4. Development of an Electrochemical Cholesterol Biosensor Based on Surface-Displayed Cho1

The freeze-dried C3 yeast cells (0.08 OD_600_ cells per strip) with different artificial electron mediators and buffer solutions, were immobilized onto the surface of screen-printed carbon or gold electrodes, respectively, to develop a whole-cell cholesterol biosensor. Like that of the glucose biosensor, hexaammineruthenium (III) chloride had more advantages as an artificial electron mediator in this cholesterol biosensor, exhibiting a lower background current (Figure 5a). Moreover, the study compared the current response of the cholesterol biosensor in phosphate buffer, hepes buffer and TES buffer. As a result, the biosensor based on 0.1 mol·L^−1^ phosphate buffer (pH 6.5) exhibited a smaller background current and a larger difference in response current from 2 to 6 mol·L^−1^ concentration of cholesterol (Figure 5b). In addition, using a gold electrode instead of a carbon paste one not only slightly improved the detection sensitivity of the cholesterol biosensor, but also contributed to the larger background current (Figure 5c).

The cholesterol biosensors were stored at room temperature and the changes in current responses towards 4 mmol·L^−1^ cholesterol were measured weekly to explore their storage stability. After 21 days of storage, the response currents of the biosensors began to gradually decrease, and the loss was less than 20% of the optimal response current (Figure 5d). It was also shown that the response currents of the biosensors increased during the first seven days of storage.

## 4. Discussion

The advantage of a surface-displayed enzyme is that the costly protein purification step can be skipped, and stability is effectively improved. In this study, two whole-cell biosensors based on YSD were developed for the detection of glucose and cholesterol. The results showed that the optimized glucose biosensor had a broad detection linear range of 20–600 mg·dL^−1^ while that of the cholesterol biosensor was 2–6 mmol·L^−1^. This indicated that the biosensors had a great potential for clinical application in the future. Additionally, the strategy presented here may provide insights on the development of other biosensors based on the cell surface display technology.

Although the cholesterol oxidase (Cho1) used in this study was reported to have outstanding activity (Appendix A), the surface-displayed Cho1 did not show the anticipated enzyme activity. This resulted to a relatively poor detection capability of the biosensor when compared with Gdh1, manifesting as a narrow linear range. Screening the highly active Cho1 mutants by directed evolution offers an available approach to problem solving [41]. It is still unclear why displaying Cho1 on the surface of yeast resulted to slow growth of the C1 strain although this may have been responsible for its low activity (Appendix A). Moreover, given that Cho1 was obtained from a bacterium (*Chromobacterium* sp. DS-1) but displayed on the surface of a fungus (*S. cerevisiae*), the mismatched protein folding system might have resulted to partial misfolding of Cho1 and subsequently low activity. Therefore, the identification of a highly active cholesterol oxidase from a fungus should be the focus of future studies.

Since yeast cells occupy more space, the amount of surface-displayed enzyme immobilized on the electrode surface was limited, leading to a lower response current compared to the purified enzymes (Figure 3e). Consequently, crushing the cells to collect pieces of the cell wall and directly immobilize them onto the surface of the electrode might solve this problem and will be the focus of future research.

## Figures and Tables

**Figure 1 biosensors-11-00013-f001:**
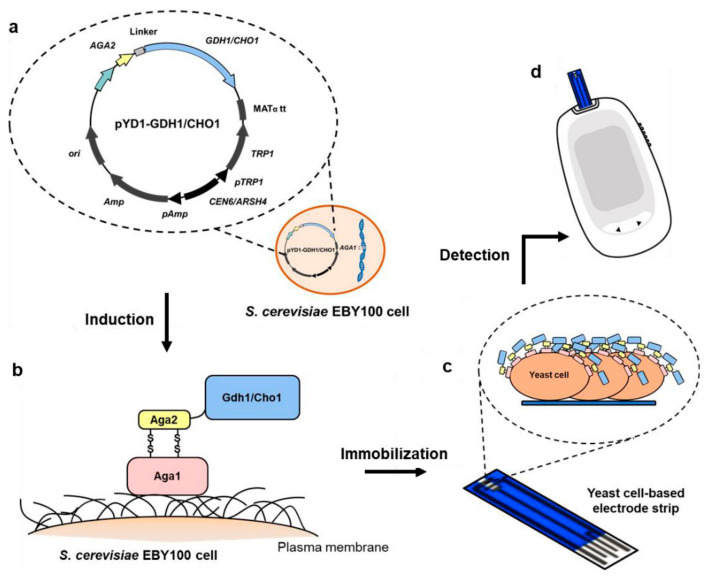
The strategy for constructing the surface-displayed glucose and cholesterol biosensors. (**a**) A map of the pYD1-GDH1 or pYD1-CHO1 vector. The enzyme Gdh1 or Cho1 (blue) was expressed as a C-terminal fusion to the Aga2 subunit of a-agglutinin (yellow) and connected by a linker (grey). (**b**) Displaying Gdh1 or Cho1 on the surface of yeast. The inducing Aga2 subunit associates with the a-agglutinin Aga1 subunit through two disulfide bonds. The Aga2-enzyme fusion protein was subsequently secreted to the extracellular space where Aga1 could be anchored to the cell wall through β-1, 6-glucan covalent linkage. (**c**) Fabrication of the whole-cell biosensors by immobilizing the YSD cells. (**d**) Electrochemical detection of glucose or cholesterol using a portable electrochemical monitor.

**Figure 2 biosensors-11-00013-f002:**
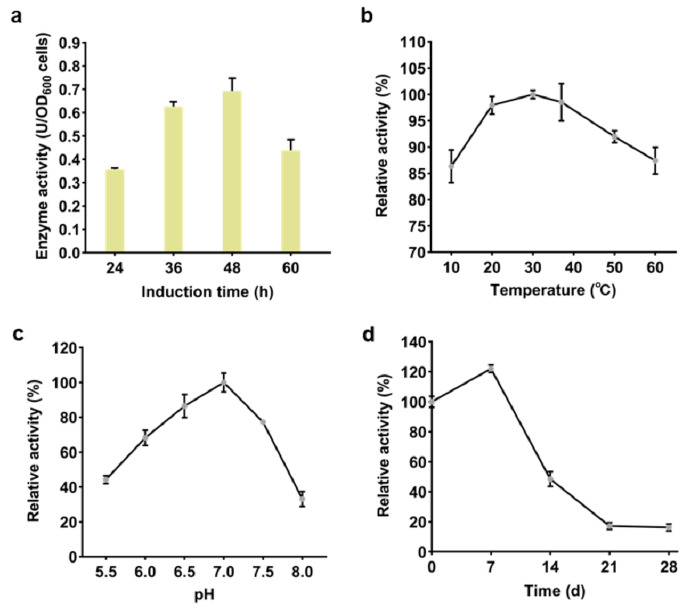
Determination of the catalytic activity of surface-displayed Gdh1. (**a**) Activity of the surface-displayed Gdh1 at different time points after galactose induction. (**b**–**d**) Effect of (**b**) temperature, (**c**) pH, and (**d**) storage time on the activity of Gdh1. Error bars indicate the SD of samples tested in triplicate.

**Figure 3 biosensors-11-00013-f003:**
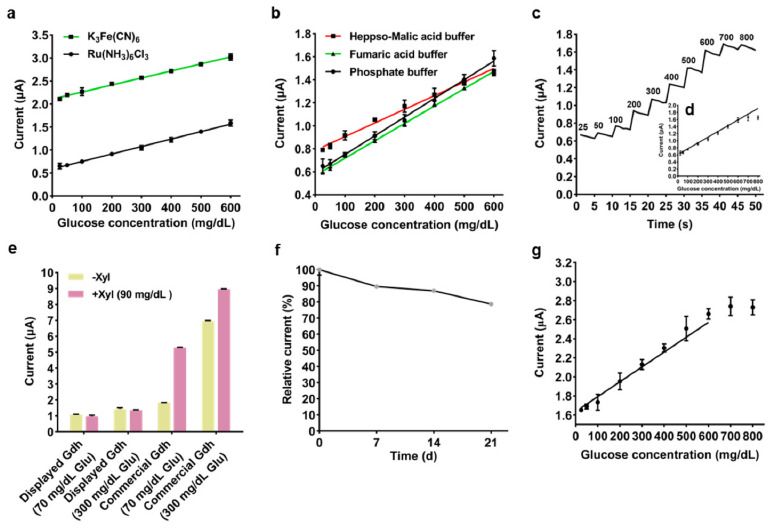
Characterization of the whole-cell glucose biosensor. Effect of (**a**) electron mediator and (**b**) buffer on the glucose biosensor. (**c**) Amperometric response for successive addition of 25, 50 and 100 mg·dL^−1^ of glucose. (**d**) The plot of linear regression. The (**e**) anti-xylose performance and (**f**) storage stability of the optimized glucose biosensor. (**g**) The response current of the optimized biosensor to glucose in whole blood samples. Electrochemical detection parameters: voltage 0.3 V, acquisition time 8.5 s. Error bars indicate the SD of samples tested in triplicate.

**Figure 4 biosensors-11-00013-f004:**
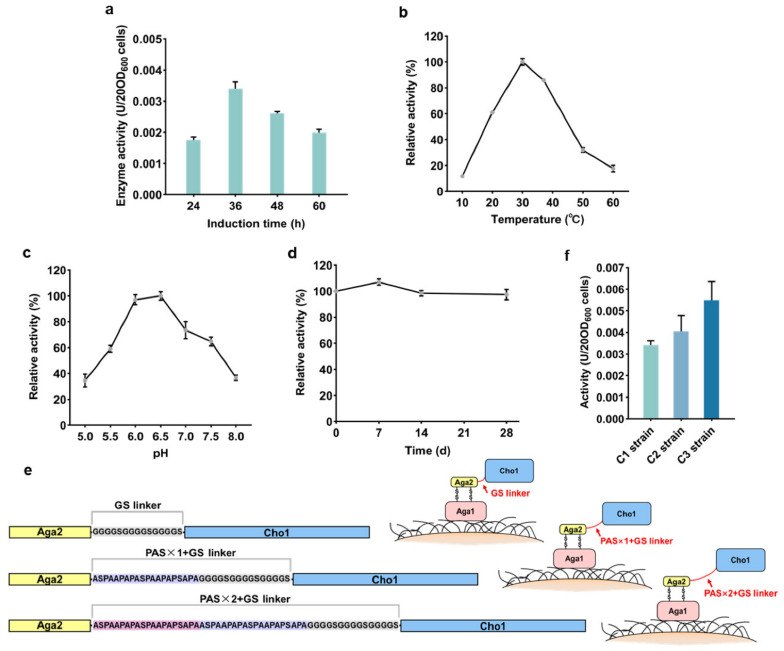
Determination of the catalytic activity of surface-displayed Cho1. (**a**) Activity of the displayed Cho1 at different time points after galactose induction. (**b**–**d**) Effect of (**b**) temperature, (**c**) pH and (**d**) storage time on the activity of Cho1. (**e**) Schematic representation of the linkers connecting Aga2 and Cho1. The original GS linker (GGGGSGGGGSGGGGS) was lengthened with one or two PAS sequences (ASPAAPAPASPAAPAPSAPA) to form the PAS × 1 + GS and PAS × 2 + GS linkers, respectively. (**f**) Activity of Cho1 with different linkers. Error bars indicate the SD of samples tested in triplicate.

**Figure 5 biosensors-11-00013-f005:**
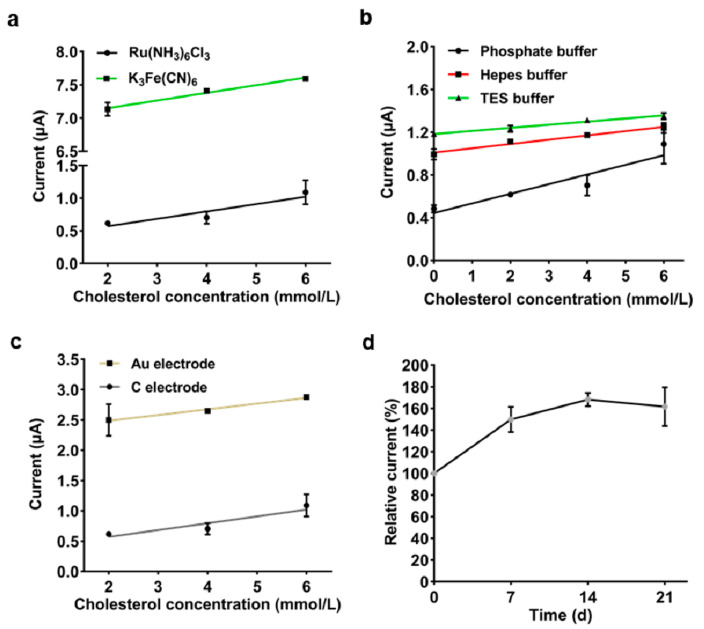
Characterization of the whole-cell cholesterol biosensor. (**a**–**c**) Effect of (**a**) electron mediator, (**b**) buffer and (**c**) electrode material on the cholesterol biosensor. (**d**) Storage stability of the optimized cholesterol biosensor. Electrochemical detection parameters: voltage 0.3 V, acquisition time 30 s. Error bars indicate the SD of samples tested in triplicate.

## Data Availability

Not applicable.

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
