# Peer review of "Display of Microbial Glucose Dehydrogenase and Cholesterol Oxidase on the Yeast Cell Surface for the Detection of Blood Biochemical Parameters"

_biosensors, 2020, doi:10.3390/bios11010013_

Round 1

Reviewer 1 Report

This paper is really interesting. Very well described and organized. I would suggest to increase the size of Figures before acceptance to be more clearly visiable.

Reviewer 2 Report

The manuscript “Display of Microbial Glucose Dehydrogenase and 2 Cholesterol Oxidase on the Yeast Cell Surface for the Detection of Blood Biochemical Parameters” by Shiyao Zhao, Dong Guo, Quanchao Zhu, Weiwang Dou and Wenjun Guan is dedicated to detection of glucose and cholesterol by enzymes displayed at the surface of yeast cells.

The article is good written and valuable however I have view doubts and comments

  1. There are no detection mechanisms presented in the manuscript. Especially as there is no information what molecule directly is responsible for the current generation that Authors measure. This is especially important as Authors are adding redox markers to the layer deposited at the screen-printed electrodes
  2. As Authors decided for electrochemical techniques to be used in Their sensors, there should be given what technique was used and its parameters should be provided. I found none of above.
  3. In electrochemical techniques the registered current (its intensity) significantly depends on the electrode surface, its conductivity, area and the same the reproducibility of its preparation. Even small differences in the electrode surface area could significantly influence the registered current intensity. In my opinion such differences are inevitable when the screen printed electrodes are used. The same, the additional differences between the electrodes surface are introduced into the assay during the deposition on the electrode surface the mixture containing the yeast cells, redox markers and etc. This in my opinion inevitably leads to significant differences between registered currents for two electrodes used exactly in the same assay. That is why any quantitative dependences between the analyte concentration and current registered are useless in electrochemical techniques when the results obtained on different electrodes are obtained. Because of the above I think that Authors should carefully review the results presented in fig. 3.

After all in my opinion the article is interesting however, because of the above doubts I recommend to reject it from publication in Biosensors (IF=3.2).

Reviewer 3 Report

The concept of surface display provides possibility of simple monitoring for several biochemicals simultaneously. However, only a limited quantity of works was focused on the application of this approach. The authors have chosen such extremely important metabolites as glucose and cholesterol as controlled compounds for their development. The prepared manuscript describes successful realization of yeast cell surface display for their monitoring.

The paper accords to basic demands of the Biosensors journal and may be published after some minor revisions:

  1. Lines 34-35. The sentence should be re-written as well as a lot of simple and rapid tests both for glucose and cholesterol are known. It will be reasonable to characterize them briefly and demonstrate the place of the proposed technique among them.
  2. Time of the assay is its important parameter that should be indicated in the Abstract.
  3. The reached linear range from 2 to 6 mmol/L is really narrow, which will limit the applicability of the sensor. The authors should discuss the possible causes of this effect and ways to overcome it.
  4. The obtained final concentration dependence of the sensor's response to glucose (Fig 3, g) should be discussed in terms of the accuracy (for example, RSD in the working range) of glucose content measurements.
  5. Fig. 5, b, curve for phosphate buffer. Taking into consideration error bars, the current values for 2 and 4 mmol/L of cholesterol are statistically indistinguishable, which narrows the actual operating range of the sensor. Please clarify this situation. The dependence should be supplemented with the current value at zero cholesterol concentration.

Round 2

Reviewer 2 Report

All issues pointed by me were successfully addressed and I recommend this article for publication.